# Executive Functions and Rapid Automatized Naming: A New Tele-Rehabilitation Approach in Children with Language and Learning Disorders

**DOI:** 10.3390/children9060822

**Published:** 2022-06-02

**Authors:** Agnese Capodieci, Marco Romano, Emanuela Castro, Maria Chiara Di Lieto, Silvia Bonetti, Silvia Spoglianti, Chiara Pecini

**Affiliations:** 1Department of Educational Sciences, Languages, Interculture, Letters and Psychology, University of Florence, 50121 Firenze, Italy; chiara.pecini@unifi.it; 2Medical Area Department, Clinical and Health Psychology School, University di Pisa, 56126 Pisa, Italy; isoladimarco@libero.it; 3IRCSS Stella Maris Foundation, 56126 Pisa, Italy; emanuela.castro@fsm.unipi.it (E.C.); mcdilieto@fsm.unipi.it (M.C.D.L.); 4Private Practice ‘Équipe Evolutiva’ in Viareggio, 55049 Viareggio, Italy; silviabonetti@hotmail.it; 5Private Practice “Paroleincerchio” in Imola, 40026 Imola, Italy; silviaspoglianti@yahoo.it

**Keywords:** tele-rehabilitation, executive function, rapid automatized naming, computerised cognitive training, working memory, learning disabilities, language disorder

## Abstract

Executive function deficits are documented in many neurodevelopmental disorders and may contribute to clinical complexity or rehabilitation resilience. The present research was primarily aimed at presenting and evaluating the feasibility and effectiveness of a telerehabilitation program used during the pandemic period. MemoRAN (Anastasis), a computerised cognitive training to improve executive control during visual-verbal integration tasks was used in a sample of 42 children (5–11 years old) with specific learning or language disorders. The MemoRAN training was based on exercises of inhibition, cognitive flexibility and updating in working memory for three months, with a frequency of approximately three sessions per week. Afterwards, a comparison between a subgroup of children using Memo-RAN and an active control group, using a tele-rehabilitation program directed on reading was conducted. Effect size analysis in pre-post measurements suggests an average effect of MemoRAN in measurements that require control processes, such as accuracy in dictation, reading, inhibition and working memory testing. Comparison with the active control group and the clinical utility implications of these types of treatment will be discussed.

## 1. Introduction

Through computerised cognitive training, tele-cognitive rehabilitation entails the recovery and/or compensation of impaired cognitive and behavioural skills to improve the patient’s quality of life in his or her family and social context. Attention, concentration, verbal and visual working memory, processing speed and inhibition are among the skills that can be improved by brain games [1]. Cognitive training programs demonstrated performance improvements in various cognitive and working memory tasks after 20 h of intervention [1,2], with sustained improvements observed over a six-month period [3,4], using an adaptive model (i.e., activity increases or decreases in difficulty depending on individual performance). Working memory deficiencies have been demonstrated to be reduced over time by using n-back tasks (those that require persons to check if a stimulus is the same as one shown earlier) that address parts of working memory [5].

Literature data show that tele-rehabilitation has been shown to be at least equivalent to traditional face-to-face rehabilitation [6,7,8,9,10] and that the tool can be effective in different settings with different patients, including intervention for the elderly [11], adults [12], children with neurological diseases [13], as well as for the treatment of motor [14], cognitive [15,16,17] and language disorders [18,19]. Within the latter category, the use of tele-rehabilitation has been shown to be effective in developmental disorders such as specific learning disorders (SLD) and language disorders (LD) [20,21,22]. SLD and LD are caused by genetic and/or neurobiological factors that modify brain function by affecting one or more cognitive processes connected to literacy. A developmental communication problem is language disorders. Language disorders affect children’s ability to communicate and understand language. They may have difficulty with written, spoken, or both languages. Children with language disorders frequently employ short or simple sentences and jumble up word order. These issues can affect literacy foundation skills such as reading, writing and/or calculating, as well as cognitive abilities such as time management, abstract logic, long- or short-term memory and attention.

According to the Diagnostic and Statistical Manual of Mental Diseases 5, SLD is defined as having literacy difficulties in critical academy skills for at least 6 months, including slow word reading, difficulty understanding what is read, spelling, written expression, number sense, mastery of numerical processes or computation and fine logic, but no intellectual or sensitive disabilities [23]. Processing of visual and audible information, which can impact reading, spelling and understanding or use of written language; planning work, with difficulty performing fine operations and following instructions; storing or retrieving information from short- or long-term memory; awkwardness or difficulty with handwriting. Attention (reaction time, information processing speed, selective or distributed attention), memory (lower memory capacity) and executive functions (shifting attention, regulation of interfering information) may all be affected in people with SLD [24]. Several studies have shown that features of oral language can be used to predict learning impairments, particularly in reading and writing [25]. The continuity between oral and written language disorders is particularly relevant in languages with regular orthography, such as Italian. Previous studies on Italian dyslexic children showed that those with a positive history for oral language impairment displayed a much more complex profile of learning difficulties in comparison with those dyslexics with a negative history for language delay or disorder. Specifically, the former was characterised by a spreader deficit in orthography and text comprehension, associated with the impairment of verbal working memory [26,27,28,29,30]. The understanding of different functional sub-groups of the SLD disorder and the identification of the main risk factors can help in choosing the intervention and in acting effective strategies to prevent the disorder.

Early intervention is critical, but it comes with a number of drawbacks, including high health-care expenses, long waiting lists and delayed therapies, as well as geographic and economic challenges for persons living distant from rehabilitation centres (e.g., rural areas). The pandemic situation of the last two years has further created the need to find different, more practical and usable solutions also for remote prevention and treatment situations.

Currently, few treatments for SLD are available, mainly to improve the reading speed and the writing accuracy, as well as the related cognitive components [31]. Current approaches to SLD focus primarily on specific processes [32,33], neglecting that at the basis of SLD there is a “multifunctional deficit model” [34], meaning that learning requires complex interactions between different specific processes and general cognitive abilities. Cognitive training for SLD, according to Zampolini and colleagues (2008, [35]), allows for continuity of treatment, reducing the time and financial demands on families and institutions. Furthermore, recent research on transparent spelling, such as that of the Italian language, has demonstrated the effectiveness of software tailored to help children and young people with SLD at home [31,36,37]. The very recent study by Maggio and colleagues (2021, [38]) shows how, even in a sample of adolescents, the use of tele-rehabilitation that combines the training of specific instrumental deficits and executive functions can provide valid support for the improvement of weak cognitive abilities in SLD. In this study, the adolescents used a 60 min telerehabilitation program for five days a week for four weeks. The results showed improvements in selective and sustained attention, shifting, interference control, memory and processing speed. 

Computerized cognitive working memory training is shown to increase academic skills including reading [39,40] and math [39,41]. Transfer effects (from working memory training to academic achievement) vary in effectiveness depending on factors such as training duration [42], starting performance [40], supervision during training [42], adding play elements to training tasks [40], motivation [43] and types of measured academic skills [39,44]. 

The use of tele-rehabilitation in SLD has been present for some years because the guidelines defined by the Consensus Conference (2011, [45]) recommend interventions in three-month cycles based on two or three meetings a week and this can often be feasible only considering the use of tele-rehabilitation. This kind of intervention has had an exponential increase with the arrival of the COVID-19 pandemic, which has forced everybody into a prolonged lockdown [46]. 

There are various types of cognitive rehabilitation software, each with its own set of features and exercises that allow the therapist to tailor the level of difficulty to the patient’s abilities, allowing for simultaneous activation of multiple cognitive areas [31,36,37].

Few studies have been conducted on the telerehabilitation of executive functions in school-age and preschool children with speech and/or SLD disorders [38,39,47].

The objective of this experimental study is to evaluate the efficacy of a tele-rehabilitation application (MemoRAN [48]), aimed at strengthening executive functions and rapid automatized naming in school-age and preschool-age children with language disorder (LD) and/or SLD. Specifically, it is a program aimed at improving the ability to inhibit the automatic response and to control the interference and the cognitive flexibility of the information kept in working memory, in a modality that requires the integration of multiple visual-verbal stimuli. The main hypothesis of the study is that this type of training can give specific improvements in terms of executive function and minor improvements in terms of academic abilities. Through the second objective, it is hypothesised to find differences in terms of improvements between a training based on the processes (MemoRAN) compared to a training based on the task (reading training). Indeed, a secondary aim of the present study is to compare the performance of rapid automatized naming, text reading (speed and accuracy) and verbal fluency of a group of school children with SLD (N = 27) who used the tele-rehabilitation program on EF, MemoRAN, with a group of children with SLD (N = 16) that used a tele-rehabilitation program on reading automatization (reading trainer, RT). 

## 2. Materials and Methods

### 2.1. Sample

For the experimentation of this tele-rehabilitation program, two groups of children were selected according to their age (preschool and school), to whom specific test protocols were administered, before and after the treatment. This is a sample collected through a multicentre study in which participants came from different centres of north and centre Italy. Foundation IRCCS Stella Maris acted as leader and participated tougher with the Research and Treatment Centre Stuttering and Voice and Language Disorders S.r.l., Fondazione Don Carlo Gnocchi ONLUS, UONPIA AO Lodi, ASST Lecco, Niguarda Hospital, CCNP San Paolo, NPI Verdello and NPI ASSTBGOVEST. A multidisciplinary team that includes a child neuropsychiatrist, a psychologist and a speech therapist performed a cognitive and learning assessment to check for a language or learning disorder. The assessment was carried out through standardised tests adapted to the age of the children and which assessed language, learning, cognitive and executive function skills.

The sample consists of 42 valid cases, of which 24 were recognised with a diagnosis of SLD (57.14%) and 18 with a diagnosis of language disorders (42.86%). Eleven belong to the group of preschool children and consequently attend preschool; the remaining 31 were primary school children (1 first, 4 second, 14 third, 9 fourth and 3 fifth grade). Characteristics of the sample in terms of age and IQ are presented in Table 1. 

Inclusion criteria to participate in the trial were: children of age from the last year of kindergarten until the last year of primary school; the presence of a specific disorder in oral language and/or reading/writing learning processes; the presence of comorbidities such as attention deficit or motor coordination disorders was not a criterion of exclusion; a neuropsychological profile suggesting the need for a strengthening of working memory skills and control of visual-verbal information.

A secondary aim of the present study is to compare the performance of a subgroup of school children with SLD (N = 27) who used the tele-rehabilitation program on EF, MemoRAN, with a group of children with SLD (N = 16) that used a tele-rehabilitation program on reading automatization (Reading Trainer, RT). Children have a mean age of M = 108 (34) and attend third (8), fourth (6) and fifth (2) grade.

### 2.2. Assessment Materials

Regarding the assessment instruments, specific protocols were developed to be administered to children before and after the MemoRAN tele-rehabilitation treatment. A distinction was made between a pre-school protocol, developed for younger children and a school protocol, which includes more advanced tests, including school reading and writing skills.

### 2.3. Preschool Protocol

A measure of cognitive control was collected, related to the cognitive abilities of children involved, through WPPSI-III [49]. This test represents the main instrument for the assessment of intelligence in preschool children from 2.6 to 7.3 years of age; it is used for the assessment of total intellectual quotient (TIQ), verbal intellectual quotient (VIQ) and performance intellectual quotient (PIQ). The battery consists of 14 subtests: 7 verbal, 5 performance and 2 processing speed subtests.

−Digit span forward (BVN 5–11 [50]): it is a task in which the child is asked to repeat, in the same order, an increasing series of numbers. −High-frequency bisyllabic word list repetition test [51]: this is a test to be performed on the PC, which allows for assessing the auditory-visual working memory. The child is presented with audios, referring to concrete objects and after listening a series of figures are presented on the screen; the child will have to indicate the figures that s/he has heard, in order of presentation. The exercise is made up of a series of 5 audios each, in which the number of objects named in each series increases progressively, starting from 2 elements; if the child indicates correctly at least 3 out of 5 trials, s/he goes on to the next series, increasing the number of named figures. The auditory-visual memory span is given by the number of objects correctly named, within a series, in which the named figures are at least 3.−Inhibition (NEPSY-II, [52]): it is a test in which shapes are presented (squares and circles or arrows in different directions) and the child must name the shapes, first correctly (conditions A—naming) and then invert them inhibiting automatic responses (condition B—inhibition) and finally following different criteria of naming (condition C—switching). At preschool age, only conditions A-B are administered, calculating time spent and errors/self-corrections made. −Verbal fluency (NEPSY-II, [52]): the child is given a semantic category or phonological cue and is asked to produce as many words as possible in one minute. −Sustained attention (AS) test (Leiter-3, [53]): it is a “barrage” test in which there are animal figures that the child must identify and cross out in 30 s; it is divided into 4 parts and the number of targets found is measured. −Ranette (BIA, [54]): it is an activity that requires auditory sustained attention skills, selective attention and inhibition of motor response. It is a go/no-go type task, in which the child must inhibit the motor act, required in the presence of a specific go sound, when the sound presented is slightly different from the previous one (no-go sound); the number of correct answers is scored. −Test of rapid automatized naming of colours and figures (RAN, [55]): the test consists of matrices of figures and colours, two matrices for each condition and the child must name them aloud, in left-right order (such as reading), noting the time taken and errors/self-corrections made. −BRIEF-P [56]: it is a questionnaire that allows evaluating in depth the EFs through parent perception, to detect the behaviour of children between 2 and 5 years; it provides scorings for 5 clinical scales: inhibition, shift, emotion regulation, working memory and planning/organisation, besides a global composite score.

### 2.4. School Protocol

Even at school age, a measure of cognitive abilities of children was collected through WISC-IV [57]. This test represents the main instrument for the assessment of intelligence in school children and adolescents from 6 to 16.11 years of age; it is used for the assessment of total intellectual quotient (TIQ), verbal comprehension index (VCI), visual perceptual reasoning index (PRI), working memory index (WMI) and processing speed index (PSI). The battery consists of 15 subtests: 5 verbal, 4 visual perceptual, 3 working memory and 3 processing speed subtests.

−Digit span forward and backward (BVN 5–11 [50]): in addition to the forward test, in which the child is asked to repeat in the same order increasing series of numbers, the backward version requested the child to repeat in reverse order with respect to the presentation.−Listening span test elementary (LSTE [58]): the test consists of a series of elementary sentences, at the lexical, syntactic and meaning level, for the assessment of working memory in children aged 8–11 years. Sentences are organised in 4 blocks, each preceded by an example. The task is to judge the semantic correctness of each sentence, i.e., whether it is true or false, and to remember the last word of each sentence heard. The elements investigated in this test are the number of last words remembered, the possible intrusions, the inversions of order in remembering the words and finally errors of judgement.−Inhibition and verbal fluency (NEPSY-II, [52]): these subtests were the same proposed for pre-schoolers. For Inhibition, in addition to conditions A-B, condition C (switching) is also administered, always examining time spent and errors/self-corrections made. −Test SD4 (PRCR-2, [59]): the test required searching a sequence of letters. It is presented as matrices of sequences of letters, progressively less spaced between them, in which the child must mark on the paper all the strings “TOC” that identifies; the clinician investigates the time taken, the errors made and the number of targets found. −Test of rapid automatized naming of figures and numbers (RAN, [55]): the same as the preschool protocol, but with figures and numbers as stimuli. −Text reading and comprehension (ALCE, [60]): text reading (text 1 in the T0 assessment and text 2 in the T1 assessment), assessing reading speed in syllables per second and error rate, together with comprehension of the text read (through ten open questions) were analysed. −Words dictation (DDE-2, [61]): a word dictation test has been included, in which the number of errors made is investigated. −BRIEF-2 [56]: like the preschool protocol, a questionnaire that allows evaluating in depth the EF through parent perception. 

### 2.5. Tele-Rehabilitation Program

MemoRAN consists of 8 different exercises, to be carried out on a PC or tablet, which involve the exposure and timed naming of stimuli, such as various figures that are presented in matrices. The program tends to exercise, singularly or in interaction, the skills of inhibition, cognitive flexibility and updating in working memory. The exercises, organised by increasing difficulty, provide specific instructions that are presented on the screen, at the beginning of each activity, based on the naming, following different rules, of a series of 5 figures (overall about 400), drawn in black and white, attributed to various libraries according to word structure and lexical complexity and length, and type of stimuli (colours, figures).

MemoRAN is part of the RIDInet platform [48], an online platform for remote rehabilitation of SLD in developmental age; it includes a series of applications that, thanks to the algorithm of self-adaptive exercises, allows a form of progressive online training adapted to the profile of each child. In detail, MemoRAN allows to perform an innovative intervention on EF, increasing the difficulty of the task in three aspects simultaneously:−the processing capacity of the system, dictated by the temporal rate at which stimuli are presented (presentation time and inter-stimuli time) and the complexity of instructions;−the integration of modalities, enhancing visual-verbal processing, going to modify the parameters of the activity, reducing or enlarging the visual and verbal load, manipulating figure size and/or word length and complexity;−the components of EF involved, thanks to the gradual progression of each exercise, which allows moving from the involvement of simpler to more complex components: first exercises work mainly on inhibition, then intervenes the working memory and finally an integration of the two EF until the introduction of cognitive flexibility.

In addition, MemoRAN is the first application of the RIDInet platform to provide, in addition to the online program for the execution of exercises, a parent app for smartphones, which requires parents to monitor the activities carried out by the child on the PC, and mark the correct or incorrect answers, communicating directly with the online program, which automatically sends a report including the number of errors, times of presentation of stimuli, types of exercises performed and parameters used, which in turn will be monitored online by the clinician. Therefore, a first session with the parent or with the adult accompanying the child during treatment is necessary to explain functioning. Regarding the timing of the treatment, the daily exercise needs to be carried out on a tablet or PC lasting 15–20 min, at least 3–4 times a week, for a period of three months. 

The theoretical principle on which MemoRAN is based is Diamond’s EF model, according to which there are three basic components, related to each other: inhibitory control, working memory and cognitive flexibility; to these are added complex components, such as planning and problem solving [43,62]. The hypothesis is that the main components to promote are working memory and inhibitory control, developed at preschool age, to which is added cognitive flexibility at school age; moreover, being core elements for school learning, it becomes essential to intervene early. Finally, another founding principle of the program is the possibility to present activities with a certain level of difficulty, novelty and diversity: some studies have shown that this postulate can be respected through the forms of tele-rehabilitation and self-adaptivity, which allow weighting the difficulties and integrating various neuropsychological elements in the activities, creating different exercises according to the child performance [63,64]. MemoRAN is developed on these theoretical lines. It consists of exercises of rapid visual naming (RAN) in which secondary activities are inserted, involving the main components of EF. The stimuli are presented with a time that starts from a specific value, identified by a first calibration exercise, presented at the beginning of the program, and accelerated gradually depending on the correctness of the naming during the temporised presentation. Time is therefore a fundamental variable, both for the capacity of visual-verbal integration, and for the ability of inhibition, updating in working memory and cognitive flexibility.

In summary, MemoRAN is composed of 8 types of exercises, set in a narrative context, which act in different ways on the various components of FE (see Table 2).

The MemoRAN session generally starts with an untimed matrix, which is necessary for the calibration phase; the following sessions will start with a calibration matrix only if the category of stimuli was changed in the previous session. Subsequently, a series of exercises of different types are proposed, presented gradually from number 1 to number 8, compatible with the duration of the session. The types of exercises are proposed as uniformly as possible, respecting the criterion of progressive appearance of the exercises based on the age of the child. 

The way the images are presented in the various exercises are changed automatically and include, as shown in Figure 1: single stimulus, progressive stimuli, anti-progressive stimuli and all visible. In this way, it is possible to work on various aspects of the control of visual information and attention: the single stimulus facilitates the left-right shift of attention and concentrates attentional focus on the presented stimulus; the progressive stimulus requires the ability to divert attention from the previous figure; the anti-progressive stimulus allows subsequent planning, simplifying the disengagement from the previous figure; all visible stimuli require a form of autonomous organisation.

The MemoRAN program is considered completed when the target time, defined for school level, is reached, on the target category of stimuli (structure and lexical complexity and length of the corresponding word); in fact, each time the target time is reached, the program moves on to the next category of stimuli and consequently more difficult for the child, e.g., moving from high lexical frequency bisyllables to high lexical frequency trisyllables, until the target is reached.

A fundamental aspect, within the competence of the clinician, is the configuration, which concerns the parameters, the objectives and the exclusions. Regarding the parameters, the clinician can set, and subsequently modify, the duration of the session, the number of naming for each exercise, whether or not to activate the pre-signal of the stimulus on the screen and whether to force the repetition of some exercises; in addition, the clinician can change some parameters that are usually blocked because self-adapted by the system, such as the exposure time of the stimuli, the size of the images and the category and complexity of the stimuli. After changing these parameters, the automatic progression can be reactivated, self-adapting the parameters based on the child’s performance. Regarding the objectives, the clinician will set the times, the size of the images and the category and complexity of stimuli to reach in the rehabilitation pathway, according to the school level. Regarding exclusions, clinicians can choose to exclude some types of exercises or categories of stimuli; for example, if certain activities are too much difficult for the child.

A final aspect to be considered concerns monitoring, which allows the clinician to check the progress of the child’s rehabilitation treatment at distance, directly from the online platform. In detail, from the monitoring screen (Figure 2) it is possible to observe along the axes the path of the child in terms of stimulus categories, errors, speed of presentation, duration of the individual daily sessions, session dates and time and accuracy for each session and its evolution. In addition, it is possible to further investigate the rehabilitation pathway, observing the parameters used, the stimuli presented, and the accuracy achieved.

### 2.6. Procedure

For both groups of children, pre-schoolers and school children, the research design has provided an initial pre-treatment assessment (T0), in which the entire age-specific protocol was administered. Then, the registration phase of the child in the RIDInet platform took place, aimed at activating the tele-rehabilitation service with the application of MemoRAN. The registration phase was managed in a clinical setting with the presence of a caregiver (usually a parent, or both), the child and the clinician. It was explained to the parent(s) what the MemoRAN tele-rehabilitation service consists of, its purpose and why it is used; the access data to the online platform were provided and the correct start and implementation of the work session were explained, as well as the functioning of the parent’s smartphone app, the various types of exercises that the child would perform, the frequency with which s/he should do them and the duration and frequency of the treatment. So, the first work session was carried out in the outpatient clinic with the clinician, to show the parent(s) the way they would have to work with their child at home.

The treatment period lasted about 3 months, at the end of which the child returned to the clinic and carried out the post-treatment evaluation (T1) with the same test protocol used at T0. It is important to emphasise that throughout the rehabilitation period, the clinician had the task of monitoring the progress of the treatment carried out by the child at home, checking the frequency and progress of the exercises, also observing any progression in terms of difficulty of the activities, due to the paradigm of self-adaptive application; in some cases, it was necessary to manually modify the parameters, also investigating the opinions and needs of the parents. 

### 2.7. Statistical Analysis

Statistical analyses were performed using the Statistical Package for Social Sciences, version 25.0 (IBM SPSS Statistics, Armonk, NY). Data were analysed in the whole sample with both descriptive and inferential statistics to test for significant changes in neuropsychological measures at T1 compared to T0. Descriptive statistics and analysis of the normality of the distribution (skewness cut-off = 2; kurtosis cut-off = 3) were carried out on all measures. Given the non-normality of RAN response time, the BRIEF questionnaire and reading accuracy measures, both parametric and nonparametric analyses, were run to analyse the results of the tests.

Due to the small sample size and the normality of most of the variables, for inferential analysis, Student’s *t*-test for paired samples was used, considering children from different ages, preschool and school children, as one group. All comparisons were planned a priori and therefore no posteriori correction procedure was used. For no parametric variables, the Wilcoxon signed rank-sum test was used. To assess the effect size, Cohen’s d was calculated for each variable, interpreting the effects according to Cohen’s d criteria [65]. Results are reported in Table 3.

Considering the difference between the performance (row scores) at the post-test minus the performance at the pre-test, the deltas (Δ) on each measure were calculated. A lower Δ corresponded to a larger effect of the training. Subsequently, the correlations between the delta and between the deltas and the score on the BRIEF questionnaire at the pre-test were analysed to investigate eventual relationships between the changes of the different tasks and if a high score on the questionnaire could correlate with a different degree of change in the tests proposed. Pearson correlations were considered for normally distributed variables; meanwhile, Spearman correlations were used for not normally distributed variables (RAN response time, BRIEF questionnaire and reading accuracy).

To investigate the effectiveness of the intervention with MemoRAN in comparison with other tele-rehabilitation interventions, the results of a sub-group of the scholar children with SLD (N = 27) who used MemoRAN in the present study were compared with results obtained from children with SLD (N = 16) using directly targeting reading intervention (reading trainer, Anastasis, for a description of the software [31]). For this aim, a multivariate analysis of variance (MANOVA) for normally distributed measures was conducted. The comparison was possible only on a few variables due to the different protocols used. To analyse the results of the rapid automatized naming, text reading (speed and errors) and verbal fluency, mixed analyses of variance with group (SLD vs. TD) as the between factor and time (pre vs. post) as the within factor were used. 

## 3. Results

Parents and children positively welcomed the tele-intervention procedure, and all children completed the MemoRAN program respecting the protocol requiring at least three times a week for three months. For some children, it was needed to integrate the auto-adaptivity of the training with parameters modification chosen manually by the clinician to make the exercise more suitable for the child’s special needs. The descriptive pre-post training measures and the results from the inferential statistics are reported in Table 3.

From the results reported in Table 3, it can be noticed a statistically significant improvement in performance on the words dictation test (DDE-2); and in decoding skills, both in speed, where speed in syllables per second (syll/s) increased by 0.23 syll/s, and in reading accuracy, where there were approximately 2% fewer errors. Conversely, no significant changes were found in text comprehension skills. Rapid lexical access skills showed a significant change in both the verbal fluency test and rapid visual naming speed. Attention abilities did not show improvement in terms of speed or number of targets found but tended to significance in the number of errors made. Inhibition skills changed significantly in terms of accuracy but not in terms of speed, where only a tendency towards significant change was observed. Considering verbal working memory tests, there were significant changes in the backward span test and the listening span test in terms of the number of words remembered, but not in the forward digit test. Analysing the BRIEF questionnaire, it was observed that the difference between the pre/post evaluation of the child’s EF, reported by the parents, is almost zero, with the mean values very close to each other.

Analysing the correlation of the deltas (Δ) of the tasks between each other emerged a negative correlation between Δ time to inhibit and Δ in reading speed (*r* = −0.486, *p* = 0.014), and between the latter and Δ in writing accuracy (*r* = −0.404, *p* = 0.045). So, as the reading speed increased, the number of write errors decreased and the response times in the inhibition tasks decreased. The Δ in writing accuracy correlates with Δ in inhibition accuracy (*r* = 0.404, *p* = 0.042) and Δ in time to inhibit negatively correlates with Δ in visual search accuracy (*r* = −0.517, *p* = 0.005). So, as the inhibition errors decrease, writing errors decrease too, and as the inhibition time decreases, errors in visual search decrease too. Finally, Δ in verbal fluency correlates with Δ in rapid automatized naming accuracy (*r* = 0.496, *p* = 0.002), showing that as the verbal fluency increased, the accuracy in rapid automatized naming increased too. Considering the correlations between deltas of the different tests and the BRIEF questionnaire emerged a positive correlation with Δ in verbal fluency (digit forward, *r* = 0.354, *p* = 0.034).

The second study’s goal was to compare, on some variables, a sub-group of children with SLD (N = 27) who used MemoRAN in the present study, with the results obtained from children with SLD (N = 16) using directly targeting reading intervention (reading trainer). A comparison in terms of age, IRP and gender show no difference between the two groups in any measure (age: *F* < 1; IRP: (*F*(1, 38) = 2.12, *p* = 0.154).

Considering the variables on which the comparisons were performed, the following results emerged. Regarding speed in text reading, a main effect of time (*F*(1, 36) = 11.40, *p* = 0.002) and the effect of interaction (*F*(1, 36) = 4.37, *p* = 0.042) were found. Children that used the RT program showed a greater improvement in syll/s in comparison with children that used MemoRAN (post-test values: M = 1.80 (0.63) vs. M = 1.55 (0.99)). Regarding accuracy in text reading, the main effect of time approaches significance, (*F*(1, 37) = 3.70, *p* = 0.062), but the interactions were not significant. 

Considering speed in rapid automatized naming (number condition), a main effect of time was found, (*F*(1, 40) = 10.57, *p* = 0.002), but no interaction effect emerged. No significant effects emerged in the case of errors. Finally, regarding verbal fluency, the main effect of time did not reach significance (*F*(1, 40) = 3.23, *p* = 0.080), and no interaction emerged. 

## 4. Discussion

The primary purpose of the research was to verify whether a tele-rehabilitation program, such as MemoRAN, could be feasible and effective in the intervention in preschool and school children, with LD and SLD, improving performance in tasks related to the abilities trained and in school skills, acting on the enhancement of EF. The objective was partially achieved as specified below. 

From the qualitative reports and the observations conducted by the clinicians, the intervention with MemoRAN resulted to be feasible, as no child dropped out of the training and parents positively welcomed the home-based interventions. Thanks to the online monitoring system, guaranteed by the tele-rehabilitation tool, it was possible to analyse the performance in the treatment period. Visual inspection of the training trajectories suggested the presence of an interaction between the differences in the child’s behavioural and attentional characteristics and the progress at the treatment. Children showing high compliance completing the exercises and carrying out the required number of sessions, families displaying collaboration skills, improved in each training session in terms of fewer errors made, faster advance in the speed of answering and achievement of a higher complexity of the exercises was observed. Conversely, the clinicians reported that children with emotional-behavioural difficulties showed a slower and less linear improvement. These aspects were partially confirmed from the positive correlation between the score on the BRIEF questionnaire and the Δ in verbal fluency; but as these results were based mainly on subjective observations, further studies may investigate the inter-subject differences in the response to intervention by structured survey and questionnaires [66]. 

For what concerns the first aim of the study, the results from the group trained by MemoRAN suggest the effect of the training on the enhancement of several measures of cognitive control and on reading and writing skills. 

In the core cognitive processes, there was a greater effect on the processes of inhibition, and working memory, compared to those of selective attention. Accordingly, also in working memory, the changes concerned the active processes of manipulation of information in memory (e.g., digit back and listening span test) rather than the passive processes of memory retention (e.g., digit forward). These results were expected as MemoRAN was directed to the core EF components, with several exercises on inhibition and updating in visual-verbal working memory, while other cognitive processes and automatized abilities were less trained. In agreement with this interpretation, it can be noted that, albeit in descriptive terms, the changes mainly concerned accuracy of response, requiring executive and cognitive control, rather than speed measurements.

For what concerns the performances in the reading and writing tests, the significant improvements were related to spelling accuracy and reading decoding speed and accuracy. Conversely, the text comprehension skills did not show significant differences between the pre and post assessment: plausibly, they require greater integration of the control processes with the instrumental skills. These profiles suggest the effect of MemoRAN on the consolidation of lexical and orthographic representations without the generalisation to semantic-contextual inference strategies. The results that emerged are in line with what is found in the literature about the efficacy of tele-rehabilitation in SLD [36,37,38,39,47]. 

At the behavioural level, it was found the absence of effects of the training on the EF behaviours detected in daily life on the BRIEF questionnaire. This result was in part expected because the training worked on the more academic and cognitive EF components while behavioural and emotional self-regulation, which usually requires contextualised and ecological training, has not been tapped. 

Finally, the analyses of correlations between Δ at the different tasks showed interesting results with a parallel decrease in errors in writing and in the inhibition test. In particular, the results showed a parallel improvement in inhibition abilities and in the efficiency of orthographic control. The decrease in the duration of the times in the inhibition test correlates with a higher reading speed and fewer errors in the visual search. Finally, as expected, and confirming the usefulness of this type of training, an improvement in the RAN leads to an improvement in verbal fluency. 

The results partly confirm what emerges in the literature regarding the improvement in the level of accuracy in learning when implementing EF. The improvement in inhibition and its correlation in terms of rapidity with speed in the literature and in accuracy with the decrease in reading and writing errors confirm the role of EF in learning.

Although the absence of a passive control group, so that it must be considered a pilot study, the calculation of the effect size, Cohen’s d, showed that the statistically significant changes mentioned above have an effect size ranging from medium to large (d > 0.50). 

Another limit was to consider children with comorbidities and analyse the sample as a whole group, due to the small sample and the continuity between language disorders and learning disabilities. In future studies, the possibilities of expanding the sample could give the opportunity of analysing separately children with or without comorbidities and with or without present or past LD.

Furtherly, a second aim of the study is to explore the specificity of the MemoRAN effects in comparison to those obtained by an auto-adaptive tele-rehabilitation training, similar in terms of procedure characteristics but working directly on reading. Albeit the comparison was possible on a few measures, the results suggest similar effects of the two trainings on decoding accuracy and rapid automatized naming, while the reading speed showed a lower change after MemoRAN than RT. These results are in line with what was expected and with the hypothesis that computerised activities aimed at implementing cognitive processes in a more general way and therefore with exercises on EF, give more generalised improvements compared to specific activities on instrumental skills (e.g., reading in this case) that show more targeted effects on trained skills than other general process skills. This result, that needs to be confirmed by future studies on larger samples, supports that a tele-intervention on the control cognitive processes can be effective and generalise the efficacy to the reading and writing skills, although to a lesser extent than an intervention working on a specific literacy skill. Thus, it can be suggested that interventions such as MemoRAN can be used in preventing the disorders before direct training could be used, and that it can be integrated with training on specific domains of learning. Within this perspective, tele-rehabilitation procedures could facilitate the simultaneous use of different types of intervention without losing the intensity of the exercises required.

## 5. Conclusions

Because SLD is a multifaceted complaint impacting cognition, it is critical to include all the cognitive functions that make up the complaint about effective recovery. It is well established that SLD people’s well-being, as well as their professional and interpersonal lives, can be harmed by a lack of early and effective recovery [67]. The use of “at home” intervention, supervised by experienced therapists, has helped to alleviate difficulties (regarding reading capacities) and lower intervention costs in the previous decade [68]. Recent studies have also demonstrated that tele-rehabilitation software is helpful in children with SLD, owing to the greater appeal of computerised courses, which allow for more cooperation by young users [31,36,37]. In particular, tele-rehabilitation programs that try to integrate multiple cognitive, verbal, visual and attention functions into a complicated exercise to support literacy are promising.

The success of tele-rehabilitation appears to be linked to the ability to simultaneously involve many cognitive processes while enforcing remote supervision when the individual is in a family setting [34,69,70]. Another essential consideration is the provision of effective and exciting assignments with competitive elements that drive challenge and improvement. As established in the field of neurorehabilitation [71,72,73], our results exhibited strong usability and interest during the training. Encouragement is highly associated with long-term advantages in recovery and quality of life [74]. This is especially true in the early phases of growth when young people must be extensively involved to urge them to stick to the training program. This is the reason why was created a system that automatically adjusted interventions to actor requests, and the stimulus parameters were updated in real time in response to the youngster’s changes, needs and preferences. Adjusting the intervention to the subject’s capacities and providing continuous feedback to the youngsters, on the other hand, develops abilities and encourages the zone of proximal development, fostering individual reading [62,75,76,77,78].

The main limitation of the study is the small sample size, which could not allow extending the results to the whole population of subjects with SLD. Considering that socioeconomic status is a variable that influences cognitive performance, another limitation is that no information about SES and parental education was presented. In addition, for a future study, it would be interesting to use a questionnaire that investigates the learning improvements observed by teachers and a follow-up assessment at 3 or 6 months to verify the maintenance of the improvements. 

In conclusion, this research demonstrates that cognitive training via tele-rehabilitation could be an effective way to help people with SLD recover. The findings imply that this technique can help these kids improve their executive functioning. Furthermore, this unique solution could boost child health care at home to match the needs of a single case, ensuring long-term care and timely rehabilitative intervention, particularly during lockdown periods. The results of this research, therefore, show the usefulness of remote rehabilitation to allow a timely and intensive intervention that consents the child to improve their academic and EF skills before these can have relapses in other spheres of their life. Indeed, it clearly emerges from the literature that children with SLD could later develop emotional and social problems if not promptly diagnosed and supported [75]. At the research practice level, further studies should investigate the integrated use of computerised programs on EF and programs on instrumental skills to verify whether the combination of these activities brings greater improvements than single or sequential use. Finally, also at the educational and scholastic level, the integration of typical learning activities could be combined with activities on inhibition processes, working memory and rapid automatized naming, to verify if learning is thus more effective and generalised for the children of the first years of primary school and in particular for children with SLD or previous LD.

## Figures and Tables

**Figure 1 children-09-00822-f001:**
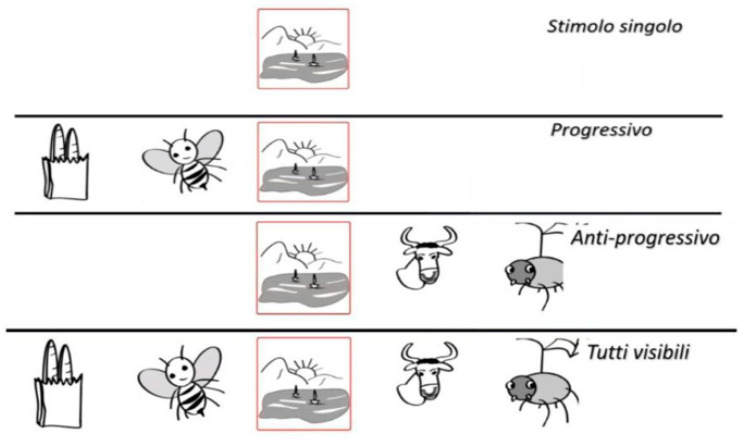
Types of stimuli presentation (in order single stimulus, progressive, anti-progressive and all visible).

**Figure 2 children-09-00822-f002:**
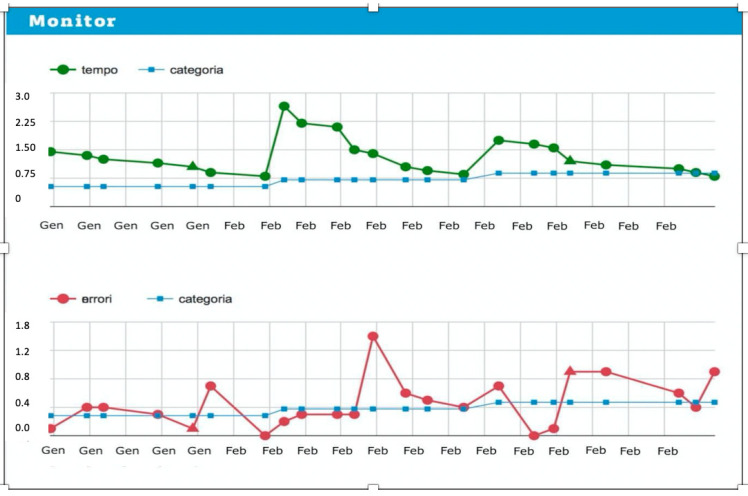
Example of monitoring screen.

**Table 1 children-09-00822-t001:** Age, perceptual reasoning Index (IRP) and intellectual quotient (IQ) of the participants, based on numerosity (N), mean (M), standard deviation (SD), maximum value (MAX) and minimum value (MIN).

	N	M (Months)	SD	MAX	MIN
AGE	42	103	42	137	59
IRP	38	99.47	12.81	132	78
IQ	37	90.43	10.53	108	70

**Table 2 children-09-00822-t002:** Types of exercises presented in MemoRAN.

Exercise 1. Inversion	This is an inhibition exercise that requires reversing the naming of two stimuli, out of five that are presented; there is a preview phase, in which the child sees which stimuli have to be named correctly and which to reverse. The parent should report errors in the application, each time the child omits or misses a name and each time s/he misses the reversal task.
Exercise 2. Cluster	In this exercise, several figures are presented or highlighted at the same time, within a cluster defined by a yellow rectangle; the child is asked to name only the stimuli highlighted by the red square (within the cluster). We can define this exercise as an activity of attentional focusing with shifting of visual and spatial attention. With each omission or error in naming, the parent will report the error on the application.
Exercise 3. Ran variable time	This task requires the child to quickly name all the stimuli s/he sees; the peculiarity is that the figures are presented with a very variable exposure and inter-stimuli time: sometimes very slow, sometimes very fast, sometimes medium. Therefore, the child will have to inhibit his own automatic rhythm and adapt to that imposed by the outside world, that is, by the advance. With each omission or error in naming, the parent will report the error on the application.
Exercise 4. Action	This is a dual-task activity, in which the child will have to name the figures as they are, as well as having to perform an action at the same time as the naming, for two specific marked stimuli. The aim is to encourage integration between a visual-verbal task and a motor control task, increasing the complexity of the processing/response mode. An error is marked both when a naming is omitted and the action is not performed (e.g., clapping).
Exercise 5. 1-back	It is a working memory exercise: the child must name the figures as they are; however, should not name the figure surrounded by the red square but the previous stimulus, that after being appeared will be replaced by a red dot; it is also important that the clinician tells the child to follow the rhythm of the visual cue. With each omission or error in 1-back naming, the parent will report the error on the application.
Exercise 6. Inversion and action	In this exercise, a complex dual task is proposed: among the five stimuli presented, two will have inverted naming, two others will be associated with an action, and the last stimulus will require simple naming as presented. Error may result from inaccuracies in naming or actions.
Exercise 7. Silence and action	This is another dual-task exercise but also includes an inhibition component: silence is required for one stimulus, silence is required for another stimulus at the same time as the tapping action on the table, and simple naming is required for the other three stimuli. The error comes from not respecting silence or respecting it for stimuli that are not required, as well as making mistakes in the execution of actions.
Exercise 8. 2-back	This exercise consists in naming the stimulus that precedes the highlighted one by two positions, keeping to the rhythm of the visual cue. If the child omits or makes a mistake in naming or does not name the figure two positions ahead, parents report the error.

**Table 3 children-09-00822-t003:** Descriptive statistics, Student’s *t*-test or Wilcox Z, significance, and Cohen’s d for protocol measures at T0 and T1.

TEST	N	M (DS) PRE	M (DS) POST	t/Z	*p*	d di Cohen
Word dictation (DDE-2–errors)	27	16.15 (9.56)	12.19(8.87)	−3.12	*p* < 0.005	−1.22 **
Alce–comprehension (correct responses)	17	6.88 (5.45)	7.41 (4.89)	−0.39	n.s.	−0.20
Alce–reading speed (syll/s)	26	1.53 (0.94)	1.76 (1.05)	−2.43	*p* < 0.05	−0.97 **
Alce–reading accuracy (errors)	26	8.78 (5.86)	6.75 (5.27)	Z = −2.172	*p* < 0.05	1.04 **
NEPSY-II–verbal fluency correct responses	37	16.72 (5.52)	17.62 (5.94)	−1.84	*p* < 0.05	−0.61 *
RAN–response time	38	125.21 (64.80)	117.95 (65.18)	Z = −2.110	*p* < 0.05	0.55 *
RAN–accuracy (errors)	37	2.36 (2.65)	2.15 (2.57)	0.61	n.s.	0.20
PRCR-2–SD4 response time	28	251.93 (89.89)	239.04 (105.05)	0.85	n.s.	0.33
PRCR-2–SD4 errors	29	8.34 (6.87)	6.62 (7.34)	1.64	*p* = 0.06	0.62 *
PRCR-2–SD4 target found	29	23.31 (6.44)	24.21 (6.77)	−0.92	n.s.	−0.35
NEPSY-II–inhibition rapidity (s)	37	59.83 (22.87)	55.69 (27.56)	1.34	*p* = 0.09	0.45
NEPSY-II –inhibition accuracy (errors)	37	4.60 (4.15)	2.66 (2.71)	3.96	*p* < 0.001	1.32 **
Digit span forward	35	4.46 (1.60)	4.66 (1.49)	−0.93	n.s.	−0.32
Digit span backward	28	3.57 (1.32)	4.07 (1.49)	−1.68	*p* = 0.05	−0.65 *
BAF-listening span test–n. words	28	10.39 (4.93)	12.29(6.38)	−2.00	*p* < 0.05	−0.77 *
BRIEF	30	18.18 (16.07)	18.20 (16.25)	Z = −0.097	n.s.	−0.03

Note: * for Cohen’s d between 0.50 and 0.90 and ** for Cohen’s d between >0.90; n.s. = not significant.

## Data Availability

The data presented in this study are available on request from the corresponding author. The data are not publicly available as the collection is still ongoing.

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
