# Peer review of "Executive Functions and Rapid Automatized Naming: A New Tele-Rehabilitation Approach in Children with Language and Learning Disorders"

_children, 2022, doi:10.3390/children9060822_

Round 1

Reviewer 1 Report

This very interesting manuscript aimed at assessing the effectiveness of a tele-rehabilitation program used during the pandemic period in preschool and school children, with LD and SLD. A second goal of the study was to compare a subgroup of children using Memo-RAN with an active control group, using a tele-rehabilitation program focused on reading. Overall, the manuscript is very well written and provides a significant counterpoint for prevailing child neuropsychology research. Though I consider that this work is highly relevant and interesting for child development researchers, I do have some suggestions for improvements in future versions of the manuscript.

- In the abstract, for greater clarity, it is suggested to present the two objectives of the study together. The wording of the second aim is confusing, how the tele-rehabilitation program is different from that of MemoRan? This is only understood by reading the method.

- The Introduction lacks a clear definition of the rationale of the second aim of the study. What are the hypotheses of the study? please clarify.

- More information regarding how children were diagnoses with LD and SLD should be provided. Who made this diagnosis?

- Considering that socioeconomic status is a variable that influences cognitive performance, it is suggested to specify whether all the children belonged to the same SES. From SES indicators, parental education is one of the most important indicators to take into account in the present study, even more considering the parent’s role regarding the evaluation of EFs (through the BRIEF) and the monitoring of their children during the tele-rehabilitation. For further clarity, it is suggested to add a table with the sociodemographic characteristics of the sample or to incorporate this data in Table 1 also including the sex of the children

- The authors mainly mention the results not providing enough explanation about them. It would be helpful if the authors could extend the discussion of findings, analyzing agreements and disagreements between the results of this study and previous related ones.

- It is suggested to add the implications of the study for clinical, educational and research practice.

Author Response

This very interesting manuscript aimed at assessing the effectiveness of a tele-rehabilitation program used during the pandemic period in preschool and school children, with LD and SLD. A second goal of the study was to compare a subgroup of children using Memo-RAN with an active control group, using a tele-rehabilitation program focused on reading. Overall, the manuscript is very well written and provides a significant counterpoint for prevailing child neuropsychology research. Though I consider that this work is highly relevant and interesting for child development researchers, I do have some suggestions for improvements in future versions of the manuscript.

- In the abstract, for greater clarity, it is suggested to present the two objectives of the study together. The wording of the second aim is confusing, how the tele-rehabilitation program is different from that of MemoRan? This is only understood by reading the method.

Thank you for the suggestion, a single objective has now been presented in the abstract.

- The Introduction lacks a clear definition of the rationale of the second aim of the study. What are the hypotheses of the study? please clarify.

An explanation about the hypotheses of the second aim of the study has now been added at the end of the introduction section.

- More information regarding how children were diagnoses with LD and SLD should be provided. Who made this diagnosis?

The information about the diagnosis center and procedure have now been added.

- Considering that socioeconomic status is a variable that influences cognitive performance, it is suggested to specify whether all the children belonged to the same SES. From SES indicators, parental education is one of the most important indicators to take into account in the present study, even more considering the parent’s role regarding the evaluation of EFs (through the BRIEF) and the monitoring of their children during the tele-rehabilitation. For further clarity, it is suggested to add a table with the sociodemographic characteristics of the sample or to incorporate this data in Table 1 also including the sex of the children

Unfortunately, we do not have information about SES and parental education and this aspect has now been added to the limitation section. Information about gender has been added.

- The authors mainly mention the results not providing enough explanation about them. It would be helpful if the authors could extend the discussion of findings, analyzing agreements and disagreements between the results of this study and previous related ones.

The conclusion has now been expanded with reference to the current literature

- It is suggested to add the implications of the study for clinical, educational and research practice.

Clinical, educational and research practice implication have now been added

Reviewer 2 Report

Thank you for this important study in highlighting the importance of treatment of learning disorders;

Authors state that “at the behavioural level, it was found the absence of effects of the training on the EF 503 behaviours detected in daily life at the BRIEF questionnaire. This result was in part ex-504 pected because the training worked on the more academic and cognitive EF components 505 while behavioural and emotional self-regulation, that usually requires contextualised and 506 ecological training, have not been tapped. “, can they explain the reasoning for using BRIEF if it looks at Behavioral components and not academic learning.

Can authors explain the “compliance in children was high”. Can authors cite other electronic programs that children are not complaint with? Can authors consider discussing the improvement in learning and if it is sustained?

If the aim is “aimed at strengthening executive functions 126 and rapid automatized naming in school-age and preschool-age children with Language 127 Disorder (LD) and/or SLD “ can authors explicitly state if the study met the aim, and if not how did choice of EF scale such as BRIEF play a role ?

Is it possible that the EF is related to other factors not discussed in this paper?

Author Response

Thank you for this important study in highlighting the importance of treatment of learning disorders;

Authors state that “at the behavioural level, it was found the absence of effects of the training on the EF 503 behaviours detected in daily life at the BRIEF questionnaire. This result was in part ex-504 pected because the training worked on the more academic and cognitive EF components 505 while behavioural and emotional self-regulation, that usually requires contextualised and 506 ecological training, have not been tapped. “, can they explain the reasoning for using BRIEF if it looks at Behavioral components and not academic learning.

Thank you for your comment. The questionnaire had been used hoping to find some changes even at behavioral level, but now it has been added in the limitation and as a suggestion for future studies to use even a questionnaire that analyzed learning improvements teachers’ point of view.

Can authors explain the “compliance in children was high”. Can authors cite other electronic programs that children are not complaint with? Can authors consider discussing the improvement in learning and if it is sustained?

What is meant by compliance has been made explicit in the text, and it has been added in the conclusions that computerized training is in itself more attractive and therefore tends to have greater compliance by young users. The improvement in terms of learning was clarified in the discussion. Unfortunately, there are no follow-up measures that allow us to verify the maintenance of improvements and this aspect was added to the limitations of the study section.

If the aim is “aimed at strengthening executive functions 126 and rapid automatized naming in school-age and preschool-age children with Language 127 Disorder (LD) and/or SLD “ can authors explicitly state if the study met the aim, and if not how did choice of EF scale such as BRIEF play a role ?

It has now been made added in the text that the objectives of the study were partially achieved and that a limitation was that of not having integrated with a questionnaire that investigated aspects of academic improvement from the point of view of teachers

Is it possible that the EF is related to other factors not discussed in this paper?

Given the complexity of executive functions, intended both the cold and more cognitive ones and the hot and more emotional ones, certainly EF are also related to other factors not taken into consideration in the context of this manuscript.

Round 2

Reviewer 1 Report

The authors have accurately addressed most of the reviewers' concerns. Being much more precise, the MS shows significant improvement.

Reviewer 2 Report

Thank you for improving it for the reader.